# Two-Point Superstring Tree Amplitudes Using the Pure Spinor Formalism

Sitender Pratap Kashyap

*Institute of Physics*
*Sachivalaya Marg, Bhubaneshwar 751005, India*

E-mail: sitender@iopb.res.in

**Abstract**

We show that the 2 point tree amplitudes in the pure spinor formalism are finite and equal to the corresponding expression in the field theories. In [1, 2], same results were presented for bosonic strings and it was mentioned they can be generalized to superstrings. The pure spinor formalism is a successfull super-Poincare covariant approach to quantizing superstrings [3]. Because the pure spinor formalism is equivalent to other superstring formalisms, we explicitly verify the above claim. We introduce a *mostly BRST exact operator* in order to achieve this.

# 1   Introduction

The two point bosonic string amplitudes in flat spacetime have been shown to be equal to the corresponding free particle expression in the quantum field theories [1, 2]. We anticipate the same to be true in the pure spinor formalism of superstrings [3]. Two point amplitudes in the bosonic strings are of the form $\frac{N}{\infty}$, where $N$ is the *supposed finite* result of direct path integral excluding the volume of unfixed CKV which is $\infty$. Thus, these amplitudes were for a very long time assumed to vanish. This conclusion is alright so long as $N$ is finite. In [1] it was noticed for the first time that $N$ is also $\infty$ and hence we need to make sense of an expression of the form $\frac{\infty}{\infty}$. It was further shown that this expression is finite and gives rise to the expected two point amplitudes. It was also mentioned that same arguments can be applied to RNS superstrings. In [2] same was repeated using the operator formalism for the bosonic strings. In the operator formalism unless three or more vertex operators are inserted, the amplitude vanishes because of the non-saturation of ghost number [4]. To get a finite result, the authors make us of a *mostly BRST exact* operator and were able to obtain the desired two point result. We also follow this approach of introducing a mostly BRST exact operator. Similar operators have been used earlier but in a different context in [5]. Before moving on, let us explain what we mean by the term mostly BRST-exact. We shall make use of an operator given by

$$V_0(z) \equiv \frac{1}{2\pi\alpha'} \int_{-\infty}^{\infty} dq \left(\lambda\gamma^0\theta\right) e^{iqX^0(z)} \tag{1}$$

The integrand of the above operator for $q \neq 0$ can be written as $[Q, \star]$ and at $q = 0$ is simply given by $(\lambda\gamma^0\theta)$. This is why it is called *mostly* BRST exact (see equation (21) for explicit form). Note that $V_0$ makes explicit use of space-time index. Thus, we must check for Lorentz invariance of the amplitude upon its insertion in the corresponding correlation function. It is important to ensure that insertion of $V_0$ does not alter the various symmetries of the amplitudes namely - super-Poincare and

conformal invariance. In appendix C we explicitly check that these symmetries are preserved[1]. Given this we shall freely use the consequences of these symmetries for making some general conclusions as we move along.

It might appear ad-hoc why to insert another operator when computing two point amplitudes. Let us try to make sense of this in the bosonic strings. For this recall that an $n-$particle S-matrix is given by [4]

$$S_{j_1 \cdots j_n} (k_1, \cdots, k_n) = \sum_{\substack{\text{compact} \\ \text{topologies}}} \int \frac{[d\phi dg]}{V_{\text{diff} \times \text{Weyl}}} e^{-S_X - \lambda \chi} \prod_{i=1}^{n} \int d^2 \sigma_i \sqrt{g(\sigma_i)} \mathcal{V}_{j_i}(k_i, \sigma_i) \qquad (2)$$

Upon carrying out a Fadeev-Popov gauge fixing, the S-matrix at the tree level becomes

$$\begin{aligned} &S_{j_1 \cdots j_n} (k_1, \cdots, k_n) \,|_{g=0} \\ &= \int \frac{[d\phi][db][dc]}{n_R} \exp\left(-S_m - S_g - \lambda\chi\right) \times \prod_{(a,i) \neq f} \int d\sigma_i^a \prod_{(a,i) \in f} c^a(\hat{\sigma}_i) \prod_{i=1}^{n} \sqrt{\hat{g}(\sigma_i)} \mathcal{V}_{j_i}(k_i, \sigma_i) \end{aligned} \qquad (3)$$

The above two equations have been taken *ipsis litteris* from [4] and hence all the symbols mean exactly same. Notice that in going from (2) to (3) the crucial thing that has happened is that the $V_{\text{diff} \times \text{Weyl}}$ in the denominator in (2) has disappeared and $c^a$ have appeared in (3). Further note that $c^a$ have appeared by fixing positions of $\kappa$ of the vertex operators, with $\kappa$ being the number of conformal killing vectors (CKV). Notice that $f$ denotes the set of vertex operator coordinates in (2) that fix *all* of the CKVs. At genus zero there are 3 CKVs. So, the above formula can be applied only to amplitudes involving 3 or more vertex operators. A **naive** application of the above to two point amplitude will give a zero result as the $c$ ghost zero modes won't get saturated. Also, we must now keep in mind that such amplitudes are not *genuine* two point functions in string theory. Thus, we need to add something extra into the *naive* two point amplitude to mimic the complete CKV fixing and getting rid of the infinite gauge volume in the denominator.

Having thus argued that we must add some addtional operators to get the correct two point amplitudes, the question that we need to address now is - what are these extra operators? On carrying out the Faddeev-Popov procedure carefully, we will get in a transparent way what these operators are. After following such a procedure, whatever we get must be the answer for string theories. It is in this way that [1] were able to clearly provide the evidence for non-zero two point amplitudes. There are however various formulations of string theories. The internal consistencies *now* require that we get non-zero two point function in each one of these formulations. As we mentioned earlier, in [2] the same issue was solved in the operator formalism. In the operator formalism, there is *now* a *new* prescription for two point amplitudes to get the desired answer. The aim of this paper is to do the same in the pure spinor formalism. [1, 2] used bosonic strings to show the non-triviality of the two point amplitudes, they mentioned that the same can be generalized to superstrings. The pure spinor formalism is one of the various equivalent formulations

---

[1]Upto BRST exact terms, which give vanishing contribution.

of superstrings. Consequently the result of this paper directly provides a proof of the claim of [1, 2] in superstrings.

Rest of the paper is organized as follows. In section 2 we first address why the two point amplitudes in the pure spinor formalism vanish on naive use of the standard prescription, then we introduce the mostly BRST operator and show that its use gives rise to the expected two point result. We end with a discussion in 3 and defer some of details and computations to the appendices.

## 2 Two point amplitudes in the pure spinor formalism

In this section we begin by addressing the problem with two point amplitudes in the pure spinor formalism. We shall consider open strings for simplicity as the generalization to other string theories is straightforward - we shall make some comments on this in the discussion section 3. The amplitude is given by [3]

$$\mathcal{A}_n = \int \prod_{j=4}^{n} dz_j \langle V_1(z_1) V_2(z_2) V_3(z_3) U_j(z_j) \rangle_{D^2} \tag{4}$$

where, $V$ are the unintegrated vertex operators and $U$ are the integrated vertex operators. We shall not present details of the pure spinor formalism here, but, mention important ingredients as we need them (see [6, 7, 8, 9] for detailed reviews). We shall be concerned only with the unintegrated vertex operators which we take to be in the plane wave basis. They are given by

$$V(z) = \hat{V} e^{ik.X} \equiv \lambda^\alpha O_\alpha e^{ik.X} \ , \qquad QV = 0 \ , \quad k^2 = -\frac{n}{\alpha'} \tag{5}$$

where, $O_\alpha$ are conformal weight $n$ fields constructed out of the world-sheet fields $\Pi^m, d_\alpha, \partial\theta^a, N^{mn}, J$ with $n$ denoting the excited level of the string and $Q$ denotes the BRST charge given by $\oint dz \, (\lambda^a d_\alpha)(z)$. Also, note that the $e^{ik.X}$ cancels the conformal weight of $O_\alpha$ so that $V$ has zero conformal weight. We should note that there is so far no analog of (2) in the pure spinor formalism which when gauge fixed gives (4)[2]. This amplitude prescription is written in analogy with (2). A justification for such a step relies on the fact that the pure spinor formalism in its non-minimal version is $N = 2$ topological strings [11] whose amplitude prescription is same as that of the bosonic strings. By now we should be able to see that a direct use of (4) to $n = 2$ is not a *genuine* two point amplitude. To get the correct two point amplitude we *must* insert some additional operator, just like bosonic string case argued previously. In order to find a cure let us look into some technical details of the pure spinor formalism.

All the non-trivial amplitudes in the pure spinor formalism can be brought to a form where there are three $\lambda$ and five $\theta$ zero modes in the corresponding correlator. We choose to normalize all the amplitudes

---

[2]Recently a gauge theory behind the pure spinor formalism was proposed in [10]. Perhaps one can arrive at the amplitude prescription using this. This has not been done so far.

with respect to the following correlator[3]

$$\langle (\lambda\gamma^m\theta)\,(\lambda\gamma^m\theta)\,(\lambda\gamma^p\theta)\,(\theta\gamma_{mnp}\theta)\rangle = 1 \tag{6}$$

For $n = 2$, a naive application of (4) gives

$$\mathcal{A}_2 = \langle V_1(z_1)V_2(z_2)\rangle_{D^2} \propto \left\langle \left(\lambda^\alpha O_\alpha^1\right)(z_1)\,\left(\lambda^\beta O_\beta^2\right)(z_2)\right\rangle_{D^2} \tag{7}$$

From the above equation it is clear that there are only two $\lambda$ present and the amplitude vanishes - in fact any amplitude with less than three unintegrated vertex operator vanishes. This suggests that the extra piece that we need must have one $\lambda^\alpha$. In the following we introduce an operator that allows us to compute the expected two point amplitude.

Let us begin by calculating the following amplitude

$$A \equiv \langle V_0(z)V_1(z_1)V_2(z_2)\rangle = \frac{1}{2\pi\alpha'}\int_{-\infty}^{\infty} dq\left\langle\left[\left(\lambda\gamma^0\theta\right)(z)e^{iqX^0(z)}\right]V_1(z_1)V_2(z_2)\right\rangle \tag{8}$$

where, $V_0$ is the operator introduced in (1) which we fix at $z$, while $V_1$ and $V_2$ are the unintegrated vertex operators (fixed at $z_1$ and $z_2$ respectively). On substituting the form of $V_i$ given in (5), we can factor the amplitude as

$$A \equiv \frac{1}{2\pi\alpha'}\int_{-\infty}^{\infty} dq\left\langle\!\!\left\langle\left(\lambda\gamma^0\theta\right)(z)\,\hat{V}_1(z_1)\hat{V}_2(z_2)\right\rangle\!\!\right\rangle\left\langle e^{iqX^0}(z)e^{ik_1.X}(z_1)e^{-ik_2.X}(z_2)\right\rangle \tag{9}$$

where, we use the notation $\langle\!\langle\cdots\rangle\!\rangle$ as a shorthand to denote that the necessary OPE have been taken among the operators inside the bracket. We have taken the momentum $k_1$ to be incoming and $k_2$ to be outgoing. The Koba-Nielsen factor of the above equation reduces to

$$\left\langle e^{iqX^0}(z)e^{ik_1.X}(z_1)e^{-ik_2.X}(z_2)\right\rangle$$
$$= iC_{D_2}^X(2\pi)^{10}\,\delta\left(q+k_1^0-k_2^0\right)\,\delta^{(9)}\left(\vec{k}_1-\vec{k}_2\right)|z-z_1|^{2\alpha'qk_1^0}|z-z_2|^{-2\alpha'qk_2^0}|z_1-z_2|^{-2\alpha'k_1.k_2} \tag{10}$$

On substituting the above result in (9), we find

$$A = \frac{i}{\alpha'}C_{D_2}^X\int_{-\infty}^{\infty} dq\,\delta\left(q+k_1^0-k_2^0\right)(2\pi)^9\,\delta^{(9)}\left(\vec{k}_1-\vec{k}_2\right)\left\langle\!\!\left\langle\left(\lambda\gamma^0\theta\right)(z)\,\hat{V}_1(z_1)\hat{V}_2(z_2)\right\rangle\!\!\right\rangle$$
$$\times|z-z_1|^{2\alpha'qk_1^0}|z-z_2|^{-2\alpha'qk_2^0}|z_1-z_2|^{-2\alpha'k_1.k_2} \tag{11}$$

From the energy delta function we have $q = k_1^0 - k_2^0 = \sqrt{\vec{k}_1^2 + m_1^2} - \sqrt{\vec{k}_2^2 + m_2^2}$. We assume that the on-shell condition is satisfied by the vertex operators $V_1$ and $V_2$ i.e. $k_i^0 = \sqrt{\vec{k}_i^2 + m_i^2}$, for $i = 1, 2$. In

---

[3]Normalizing this correlation function is sufficient since there is only one scalar present in tensor product of three $\lambda$ and five $\theta$.

the case where $V_1$ and $V_2$ represent vertex operators of strings with equal masses i.e. $m_1^2 = m_2^2 = m^2$, then $q = 0$ because of the space Dirac-delta function. In this case on performing the $q$ integral we get,

$$A = \frac{i}{\alpha'} C_{D_2}^X (2\pi)^9 \, \delta^{(9)} \, \delta^{(9)} (\vec{k}_1 - \vec{k}_2) \left\langle\!\left\langle \left(\lambda\gamma^0\theta\right) \hat{V}_1(z_1)\hat{V}_2(z_2)\right\rangle\!\right\rangle |z_1 - z_2|^{-2\alpha'm^2} \tag{12}$$

Let us next consider the case when the masses are unequal. This means $q \neq 0$ and the integrand of $V_0$ is BRST exact (see the appendix A). In this case we could have at the very beginning of the computation substituted the said non-zero value of $q$ in the $e^{iqX^0}$ and replaced the expression $(\lambda\gamma^0\theta)e^{iqX^0}$ with a BRST exact piece (see 21). Consequently for $m_1^2 \neq m_2^2$ the amplitude vanishes on the account that $QV_1 = 0, QV_2 = 0$. Hence, we can combine both cases by writing

$$A = \frac{i}{\alpha'} C_{D_2}^X (2\pi)^9 \, \delta^{(9)} (\vec{k}_1 - \vec{k}_2) \left\langle\!\left\langle \left(\lambda\gamma^0\theta\right) (z)\hat{V}_1(z_1)\hat{V}_2(z_2)\right\rangle\!\right\rangle |z_1 - z_2|^{-2\alpha'm_1^2} \delta_{m_1,m_2} \tag{13}$$

We see that there is a factor of $|z_{12}|^{-2\alpha m_1^2}$, so it appears things might not be well. However, there is an additional factor of pure spinor superspace present namely $\left\langle\!\left\langle \left(\lambda\gamma^0\theta\right) \hat{V}_1(z_1)\hat{V}_2(z_2)\right\rangle\!\right\rangle$. Recall that at $n^{th}$ level of superstring we have $(\text{mass})^2 = \frac{n}{\alpha'}$ and also that the conformal dimensions of $\hat{V}_i$ for $i = 1, 2$ are $n$. Note that $(\lambda\gamma^0\theta)$ has conformal weight zero. Thus, upon using the standard result of a three point function in a CFT, we find

$$\left\langle\!\left\langle \left(\lambda\gamma^0\theta\right) (z) \hat{V}_1(z_1)\hat{V}_2(z_2)\right\rangle\!\right\rangle \propto |z_{12}|^{2n} = |z_{12}|^{2\alpha'm^2} \tag{14}$$

which, cancels the coordinate dependence coming from the Koba-Nielsen factor. Thus, the amplitude is coordinate invariant.

Let us now compare the result we have found to the answer we have in field theory (in $D$ dimensions), where

$$\mathcal{A}_2 = 2k^0 (2\pi)^{D-1} \delta^{D-1} (\vec{k}_1 - \vec{k}_2), \qquad k^0 \equiv \sqrt{m^2 + \vec{k}^2} \tag{15}$$

Our answer in (13) doesn't seem to have any $k^0$. We argue that it is present and comes from $\langle\!\langle \cdots \rangle\!\rangle$. To see this note that we must have

$$\left\langle\!\left\langle \left(\lambda\gamma^0\theta\right) (z) \hat{V}_1(z_1)\hat{V}_2(z_2)\right\rangle\!\right\rangle = f^0(\epsilon_1, \epsilon_2; k) \tag{16}$$

i.e. the above correlator must at the end be expressible in terms of polarizations $\epsilon_i$ and momentum $k$ of the state represented by vertex operators $V_1$ and $V_2$. Also, note that on the r.h.s, we must have a $0$ index (indicated by $f^0$). Since our theory is supersymmetric, giving the argument for purely bosonic states is sufficient. The bosonic states will have their polarizations given by Lorentz vector indices. Further note that polarizations satisfy $k_m\epsilon_i^{m\cdots} = 0$. Now, suppose that $0$ index is supplied by $\epsilon_1$. Then for non-zero answer, can contract rest of the indices of $\epsilon_1$ with only $\epsilon_2$. But, this leaves a free index on $\epsilon_2$ which must

be contracted by $k$, giving a vanishing contribution. Thus, there is a unique choice - the polarization tensors contract among themselves and $0$ index is supplied by $k^0$. In appendix B we explicitly verify this for all states at massless level. Thus, we find[4]

$$\left\langle\left\langle \left(\lambda\gamma^0\theta\right)\hat{V}_1(z_1)\hat{V}_2(z_2)\right\rangle\right\rangle \propto k^0\,\delta_{jj'} \tag{17}$$

where, we have used $j$ and $j'$ in $\delta_{jj'}$ to distinguish between states with degenerate masses like gluon and gluino. Hence, the final result is

$$A \propto (2\pi)^9\,\delta^{(9)}\!\left(\vec{k}_1 - \vec{k}_2\right)k^0\,\delta_{m_1,m_2}\,\delta_{jj'} \tag{18}$$

upto a proportionality constant. In appendix B we explicitly verify the above result for some amplitudes. Thus, the two point amplitudes in the pure spinor formalism using operator $V_0$ behave as is expected.

# 3 Discussion

The non-vanishing of two point amplitudes is required for various consistencies - see [12] for a discussion. We showed that by making use of an additional set of mostly BRST-exact states we can get non-zero two point tree amplitudes in the pure spinor formalism in open strings. Generalization to the closed strings is straight-forward in the pure spinor formalism. For the closed strings, we just need to add a right sector of what we have done so far. Generalization to Heterotic strings can be done by a hybrid of [1, 2] and what we did above. For the supersymmetric sector we use pure spinor prescription presented in this work and augment it with the derivation of the bosonic sector as in [1, 2].

To conclude we essentially have defined what we mean by a two point amplitude in the pure spinor formalism. We do not however know from a fundamental point of view why the additional vertex operator is of this form[5]. Nonetheless, it is easy to explain the role of the various ingredients that went into $V_0$. The integral over $q$ along with the choice of $X^0$ helped get rid of the energy delta function. The insertion of an extra $\lambda$ unarguably is a must. This insertion of $\lambda$ must appear as a worldsheet scalar - this essentially leaves us with a unique choice of contraction with $\theta$ with a $\gamma$ matrix i.e. $(\lambda\gamma^m\theta)$. Also, $m = 0$ appears naturally once we isolate $X^0$ for taking care of the energy delta function. We end this work by saying that it is worth exploring for a fundamental origin of the mostly BRST exact operator we used in this work, perhaps by making use of the gauge invariant action presented in [10] (see also [13, 14] which gave important insights that lead to [10]) . This investigation we leave for future work.

**Acknowledgments:** I wish to thank Biswajit Das for some discussions and Ashoke Sen and Mritunjay

---

[4]There are other factors containing contribution of non-zero modes of various worldsheet fields, normalization of polarizations and pure spinor superspace computations. These are all non-zero.

[5]Perhaps these are related to the zero momentum states as pointed out by Renann Lipinski Jusinskas. We note that the additional operators in [1, 2] too resemble zero momentum states. The understanding of their precise role needs more work.

Verma for providing some useful comments on the draft. I am indebted to Renann Lipinski Jusinskas for many insightful discussions at the initial stages this work. I am thankful to Institute of Physics, Bhubaneshwar for generously providing a three month extension beyond the usual term of my post-doctoral tenure, during the pandemic due to CoVid-19.

## A $\quad V_0$ as a mostly BRST exact operator

In this appendix we show that the $V_0$ introduced in the main body is a mostly BRST-exact operator. Let us begin by noticing

$$\left[Q, e^{iqX^0(z)}\right] = \frac{\alpha'}{2} \oint_z dw \lambda^\alpha(w) \left[\frac{D_\alpha e^{iqX^0(z)}}{w-z} + \cdots\right] \tag{19}$$

where, we used the standard OPE $d_\alpha(z)V(w) \simeq \frac{\alpha'}{2}\frac{D_\alpha V}{z-w}$. On recalling that $D_\alpha = \partial_\alpha + (\gamma^m\theta)_\alpha \partial_m$, we find that

$$\left[Q, e^{iqX^0(z)}\right] = \frac{iq\alpha'}{2} \left(\lambda\gamma^0\theta\right) e^{iqX^0} \tag{20}$$

Hence, for $q \neq 0$ we have

$$\left(\lambda\gamma^0\theta\right) e^{iqX^0} = -\frac{1}{q} \left[Q, \left(\frac{2i}{\alpha'}e^{iqX^0(z)}\right)\right] \tag{21}$$

showing that the integrand of $V_0$ is BRST-exact for $q \neq 0$ and thus $V_0$ is mostly BRST exact.

## B $\quad$ Some explicit examples

In this appendix we explicitly calculate the two point massless open superstring amplitudes (on a disk) with the new prescription given equation (8) in this paper. The goal is to substantiate the claim in (17) by providing some explicit examples. For this we essentially need to compute the $\left\langle\left\langle \left(\lambda\gamma^0\theta\right) \hat{V}_1\hat{V}_2 \right\rangle\right\rangle$ where $\hat{V}_i = \lambda^\alpha A_{i\alpha}$ and show it is proportional to $k^0$. For the massless case the task is trivial as everything inside the bracket is conformal weight zero and hence there are no non-trivial OPEs. Consequently we can directly use the pure spinor-superspace method to perform the computation. The relevant theta expansion is given by (we follow the notation and conventions used in [15])

$$A_\alpha = a_m \left(\gamma^m\right)_\alpha - \frac{2}{3} \left(\gamma^m\theta\right)_\alpha \left(\theta\gamma_m\chi\right) + \cdots \tag{22}$$

where, we have not shown the higher order $\theta$ terms as they will not be required. Also, $a_m$ represents the gluon field and $\chi^\alpha$ the gluino field. Consequently we find

$$\left\langle\left\langle \left(\lambda\gamma^0\theta\right)\hat{V}_1\hat{V}_2\right\rangle\right\rangle = \left\langle\left(\lambda\gamma^0\theta\right)\left(\lambda^\alpha A_{1\alpha}\right)\left(\lambda^\alpha A_{2\alpha}\right)\right\rangle = \frac{i}{180}k^0\left(a_r^m a_{r'm}\right) + \frac{1}{360}\left(\chi_s\gamma^0\chi_{s'}\right) \tag{23}$$

where, $r, r'$ and $s, s'$ denote polarizations and helicities of gluons and gluinos respectively. We made use of the following pure spinor superspace identities for performing the gluon calculation

$$\left\langle\left(\lambda\theta\gamma^m\rho\right)\left(\lambda\theta\gamma^n\rho\right)\left(\lambda\theta\gamma^p\rho\right)\left(\lambda\theta\gamma_{stu}\rho\right)\right\rangle = \frac{1}{120}\delta_{stu}^{mnp} \tag{24}$$

and

$$\left\langle(\lambda\gamma^u\theta)(\theta\gamma_{fgh}\theta)(\theta\gamma_{jkl}\theta)(\lambda\gamma_{mnpqr}\lambda)\right\rangle$$

$$= -\frac{4}{35}\left[\delta_{[j}^{[m}\delta_k^n\delta_{l]}^p\delta_{[f}^q\delta_g^q\delta_{h]}^{r]}\delta_h^u + \delta_{[f}^{[m}\delta_g^n\delta_{h]}^p\delta_{[j}^q\delta_k^q\delta_{l]}^{r]}\delta_l^u - \frac{1}{2}\delta_{[j}^{[m}\delta_k^n\eta_{l][f}\delta_g^p\delta_{h]}^q\eta^{r]u} - \frac{1}{2}\delta_{[f}^{[m}\delta_g^n\eta_{h][j}\delta_k^p\delta_{l]}^q\eta^{r]u}\right]$$

$$-\frac{1}{1050}\epsilon^{mnpqr}{}_{abcde}\left[\delta_{[j}^{[a}\delta_k^b\delta_{l]}^c\delta_{[f}^d\delta_g^e\delta_{h]}^u + \delta_{[f}^{[a}\delta_g^b\delta_{h]}^c\delta_{[j}^d\delta_k^e\delta_{l]}^u - \frac{1}{2}\delta_{[j}^{[a}\delta_k^b\eta_{l][f}\delta_g^c\delta_{h]}^d\eta^{e]u} - \frac{1}{2}\delta_{[f}^{[a}\delta_g^b\eta_{h][j}\delta_k^c\delta_{l]}^d\eta^{e]u}\right] \tag{25}$$

for gluino calculation[6]. We further note that the polarizations are normalized (see for example [18])

$$a_r^m a_{r'm} = \delta_{rr'}, \quad \left(\chi_s\gamma^0\chi_{s'}\right) = k^0\delta_{ss'} \tag{26}$$

Thus, we find that

$$\left\langle\left\langle\left(\lambda\gamma^0\theta\right)\hat{V}_{1j}\hat{V}_{2j'}\right\rangle\right\rangle \propto k^0\delta_{jj'}\delta_{ss'} \tag{27}$$

where, $j, j'$ stand for the particle species and $s, s'$ denote the corresponding polarizations. We note that we get the same relative factor in gluon and gluino amplitude as in [15] and that the gluon-gluino amplitudes vanish automatically as we expect them to. For the massive case the computations can be repeated using [15, 19], though somewhat involved.

# C  Some consistency checks for $V_0$ insertion

In this appendix we verify that the insertion of $V_0$ does not spoil the super-Poincare and conformal invariance. First note that the expression of $V_0$ is not Lorentz covariant as it isolates $0th$ spacetime component. Superstring theory in the flat background is super-Poincare invariant. So, let us first see if

---

[6]We acknowledge the use of [16, 17] for performing the calculations.

the expression that we have arrived has this symmetry. Notice that

$$\delta \langle V_0 V_1 V_2 \rangle = \langle \delta V_0 V_1 V_2 \rangle + \underbrace{\langle V_0 \delta V_1 V_2 \rangle + \langle V_0 V_1 \delta V_2 \rangle}_{\text{automatically invariant}} = \langle \delta V_0 V_1 V_2 \rangle \tag{28}$$

where, $\delta$ denotes the change after application of some symmetry transformation. The vertex operators $V_1$ and $V_2$ are by definition invariant under $\delta$ and hence we only need to evaluate $\delta V_0$. In order to facilitate the discussion, let us note that we can write $V_0$ as

$$V_0 = \int_{-\infty}^{\infty} \frac{dq}{i\pi\alpha'^2} \frac{1}{q} \left[ Q, e^{iqX^0} \right] \tag{29}$$

Now, we can write the variations as (on noticing that $\delta Q = 0$ )

$$\delta V_0 = \int_{-\infty}^{\infty} \frac{dq}{i\pi\alpha'^2} \frac{1}{q} \left[ Q, \delta e^{iqX^0} \right] \tag{30}$$

Let us now consider all the transformations one by one. To distinguish one transformation from other, we shall provide a subscript on $\delta$. Under translations

$$X^m \rightarrow X^m + a^m \implies \delta_a X^m = a^m \implies \delta_a e^{iqX^0} = iqa^0 e^{iqX^0} \tag{31}$$

Thus, we see that

$$\delta_a V_0 = \int_{\infty}^{\infty} \frac{dq}{i\pi\alpha'^2} \frac{1}{q} \left[ Q, iqa^0 e^{iqX^0} \right] = \left[ Q, \int_{\infty}^{\infty} \frac{dq}{\pi\alpha'^2} a^0 e^{iqX^0} \right] \tag{32}$$

On making use of $QV_1 = 0 = QV_2$, we can easily see that the two point amplitude is translationally invariant. Similarly, under Lorentz transformations

$$X^m \rightarrow X^m + \Lambda^m_{\ n} X^n \implies \delta_\Lambda V_0 = \left[ Q, \int_{\infty}^{\infty} \frac{dq}{\pi\alpha'^2} \Lambda^0_m X^m e^{iqX^0} \right] \tag{33}$$

and under supersymmetry transformation

$$X^m \rightarrow X^m + (\eta\gamma^m\theta) \implies \delta_\eta V_0 = \left[ Q, \int_{\infty}^{\infty} \frac{dq}{\pi\alpha'^2} (\eta\gamma^m\theta) e^{iqX^0} \right] \tag{34}$$

Thus, we see that the amplitude is super-Poincare invariant. Finally under conformal transformations $z \rightarrow z + \epsilon(z)z \implies \delta X^m = \epsilon(z)\partial X^m(z)$, we have

$$\delta_\epsilon V_0 = \left[ Q, \int_{\infty}^{\infty} \frac{dq}{\pi\alpha'^2} \epsilon(z) e^{iqX^0} \right] \tag{35}$$

showing that the two point amplitude is conformally invariant.

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
