# Peer review of "Two-Point Superstring Tree Amplitudes Using the Pure Spinor Formalism"

_SciPost Physics Core_

## Round 3 · Referee Report · Zhengwen Liu (Referee 1) · 2024-6-1

Strengths

  1. Following recent developments in two-string amplitudes, this paper introduces for the first time a mostly BRST exact operator within the framework of the pure spinor formalism, and discusses two-point tree-level superstring amplitudes in detail.

  2. The paper is clearly presented and includes detailed derivations and explanations.

Weaknesses

The level of originality is somewhat limited: on one hand, two-point bosonic string amplitudes have already been discussed in the literature; on the other hand, the idea of introducing an additional operator has also been discussed in similar studies.

Report

This paper presents a prescription for computing two-point tree-level scattering amplitudes of superstrings by introducing a new mostly BRST exact operator. The paper provides ample detail in derivations and explanations. Upon review, the results are found to be reasonably reliable. Although there are some minor writing issues, the paper is overall well-written. I believe this manuscript meets the standards of SciPost Physics Core and recommend its acceptance.

Requested changes

  1. The author mentiones in the discussion section, "The non-vanishing of two point amplitudes is required for various consistencies". It would be beneficial to include a comprehensive discussion on how the results of this work address this issue, specifically by elaborating on the correctness and consistency of the two-point amplitudes presented in eq. (16).

  2. Below eq.(3), a description should be provided for the pure spinor $\lambda^\alpha$.

  3. On page 4: clarity would be improved by revising "The operator (1) fulfills these requirements" to "The operator proposed in this work in (1) fulfills these requirements".

  4. Below the first equation on page 5: it would be useful to clarify whether the 1st and 3rd terms vanish individually or their combination vanishes.

  5. Below the first equation on page 5: a brief description of $\epsilon$ should be given.

  6. In the line above eq.(9): "expression" would be more appropriate than "equations."

  7. From eq.(11) to eq.(12): it would be better to unify notations $z_{12}$ and $z_1{-}z_2$.

  8. Overall, the paper is well-written. However, a number of grammatical errors need to be corrected. For example: -- "it was argued that their analysis will be carry", "be" should be removed. -- "the numerator of the corresponding path integral in finite", "in" should be "is". We do not provide a complete list, but encourage the author to run a thorough grammatical check.

Recommendation

Publish (meets expectations and criteria for this Journal)

---

## Round 3 · Referee Report · Anonymous (Referee 2) · 2024-6-18

Strengths

1, Clear resolution of the vanishing amplitude problem. 2, Rigorous mathematical formulation and validation of results. 3, Comprehensive review of the pure spinor formalism, aiding both new and experienced researchers.

Weaknesses

1, The paper's high level of mathematical complexity may be challenging for readers not deeply familiar with the pure spinor formalism. 2, The specific focus on two-point amplitudes may limit immediate broader applications. As a follow-up work to Ref.[1,2], we expect more results in a new paper.

Report

The paper provides an approach to calculating two-point tree amplitudes in the pure spinor formalism for superstring theory. Traditional methods yielded vanishing results, but the author introduces a mostly BRST exact operator that ensures non-zero amplitudes for massive external states consistent with field theory. The work verifies Lorentz invariance and maintains other essential symmetries.

Requested changes

1, In the abstract, "In [1, 2], same results ..." -> "In [1, 2], the same results ..." .

2, In the first paragraph of the introduction section, " will be carry..." -> " will be carried..."

3, In the next paragraph, "This understanding however relies on the assumption that the numerator of the corresponding path integral in finite." -> "This understanding however relies on the assumption that the numerator of the corresponding path integral is finite."

4, There are several grammatical errors throughout the paper. We recommend the author address these issues before the journal considers it for formal acceptance.

5, There are numerous results for tree-level high-point massless superstring amplitudes, including those with an arbitrary number of external states. In contrast, this paper focuses on two-point massive superstring amplitudes. It would be beneficial if the author could explore the potential relationship between these areas in the discussion section.

Recommendation

Publish (meets expectations and criteria for this Journal)

---

## Editorial Decision

resubmitted